

# The effect of personality measurement conditions on spontaneous swimming behavior in the pale chub *Zacco platypus* (Cyprinidae)

Shi-Jian Fu

Laboratory of Evolutionary Physiology and Behavior, Chongqing Key Laboratory of Animal Biology, Chongqing Normal University, Chongqing, China

## ABSTRACT

Studies on personality have revealed that some personality traits are strongly correlated; thus, researchers may be able to acquire data for variables related to different personality traits from one measurement. Therefore, the aim of the present study was to test whether spontaneous movement traits used in fish personality measurements are correlated or vary among different contexts in a common Chinese cyprinid fish, the pale chub (*Zacco platypus*, Cyprinidae). The median swimming speed, percent time spent moving and median turning rate were measured in a boldness context (with a shelter available), then in an exploration context (with a novel object nearby) and finally in a control context (i.e., with no shelter or novel object). The median swimming speed, percent time spent moving, and median turning rate all showed positive correlations between the control and the other two contexts, which suggests that future studies might use spontaneous swimming variables measured in exploration or boldness contexts to avoid the need to carry out a separate activity test. Further analysis comparing the distance to and latency to explore the novel object between the exploration context (with the novel object present) and control context (with an imaginary object at the same position) showed that the amount of time it took for the fish to first reach the object for exploration was significantly shorter in an exploration context than in a control context. This suggests that latency to explore might be useful as a variable indicating exploration in the pale chub in the future.

Corresponding author
Shi-Jian Fu, shijianfu9@cqnu.edu.cn

## INTRODUCTION

Animal personality is measured as a consistent difference among individuals in behaviors such as boldness, exploration and activity (*Sih, Bell & Johnson, 2004*; *Bell, Hankison & Laskowski, 2009*; *Réale et al., 2010*; *Mazué & Godin, 2015*; *Jolles et al., 2019*). Personality has been assumed to have large fitness consequences and a wide range of ecological and evolutionary implications (*Smith & Blumstein, 2008*; *Réale et al., 2010*; *Sih et al., 2012*); hence, it has attracted much attention recently (*Jolles et al., 2017*; *Tang & Fu, 2019*). The tendency to leave a refuge and search for an open environment is traditionally referred to as

boldness (*Burns, 2008*; *Jolles et al., 2017*), and studies have used latency to leave the shelter and percentage of time (or activity) outside the shelter as indicators of this characteristic (*Brown & Irving, 2014*; *Bevan et al., 2018*; *Tang, Wu & Fu, 2018*). The novel object test has been widely used in personality research to assess exploratory behavior (i.e., curiosity towards novelty) and neophobia (i.e., fear of novelty) (*Galhardo, Vitorino & Oliveira, 2012*). Exploration is usually evaluated in terms of the distance or latency associated with the inspection of a novel object (*Liu & Fu, 2017*). For activity measurements, fish biologists usually use the characteristics of spontaneous activities (movements without external stimulus) in an arena, for example, the median swimming speed and percent time moving during a given period (*Brown & Irving, 2014*; *Liu & Fu, 2017*). It has long been recognized that many animals exhibit a so-called 'behavioral syndromes', as bold individuals are typically more exploratory and active than other individuals (*Sih, Bell & Johnson, 2004*; *Sih et al., 2012*; *Martins & Bhat, 2014*); i.e., personality traits are highly correlated, at least with regard to boldness, exploration and activity. A study involving principal component analysis also suggested that most variables can be reduced to a so-called 'activity component', which is distinct from a 'sociability component' (*Tang, 2019*). If the variables are correlated across different personality contexts, researchers may be able to acquire data for variables related to different personality traits from one measurement. For example, one could measure the activity variables mentioned above from spontaneous movement trajectories of fish being tested in a boldness or an exploration context. Thus, the present study aimed to determine whether swimming behavior traits were correlated within individuals across different personality contexts. The contexts included a boldness context with a shelter available, an exploration context with a novel object nearby and an activity context with no novel object or shelter. I tested the correlations between the activity (i.e., control) context and the other two contexts but did not test the correlation between the boldness and exploration contexts because the test conditions were not strictly controlled between these two contexts; i.e., in the boldness context, there was a shelter nearby but no novel object, whereas in the exploration context, there was a novel object but no shelter nearby.

To achieve these goals, I selected the pale chub (*Zacco platypus*), a common small cyprinid fish species distributed across East Asia, as an experimental model. This omnivorous species often forms shoals in open water but may also be found solitarily, refuging in plants or behind stones when foraging or avoiding predators. I recorded videos of individual fish in different personality contexts with a webcam. The variables I selected for comparison were median swimming speed, percent time spent moving, and median turning rate of the body centroid (which usually serves as an indicator of exploration tendency but is also strongly correlated with activity) (*Couzin et al., 2011*; *Ioannou, Singh & Couzin, 2015*; *Sumpter et al., 2018*).

To further test whether the novel object used in the present study was a sufficient stimulus for exploration in the pale chub, I conducted an additional analysis comparing the mean distance to the novel object and the latency to explore the novel object in the exploration context with similar measures in relation to a virtual object in the activity context. The aim of this additional analysis was to test whether the novel object was an adequate stimulus or whether the correlation between variables of activity and exploration was an artifact of the
measurement protocols. For example, more active fish (individuals that spend more time moving and with a faster speed) might by chance take less time to reach a novel object independent of their exploration tendency. Because no investigation has been conducted to test this issue, it might provide a useful result for studies of exploration measurements in the future.

## MATERIAL AND METHODS

### Source of the fish and their care

A total of 80 juvenile pale chubs (body mass: 6.42 $\pm$ 1.26 g; mean $\pm$ S.D.) were captured by local fishers from a tributary of the Wujiang River (29°24′37″N, 107°31′55″E, Wulong County, Chongqing city). All fish were held in a 250 L recirculating system with aerated water. Twenty percent of the water was exchanged daily with fresh water. The water temperature was maintained at 25 $\pm$ 1 °C. The photoperiod was 12 h light:12 h dark. The pale chubs were hand-fed to satiation once daily (at 8:00 am) for 4 weeks with tubifex. The feces and uneaten food were removed with a siphon at 9:00 am. The dissolved oxygen level was kept above 90% saturation. After 4 weeks of acclimation, experimental fish were tagged intraperitoneally with passive integrated transponder (PIT) tags under anesthesia by neutralized tricaine methane sulfonate (MS222, 50 mg L$^{-1}$) and allowed to recover for 2 weeks. All individuals were used to measure personality.

The present study was authorized by the Animal Care and Use Committee of the Key Laboratory of Animal Biology of Chongqing (permit number: Zhao-20161122-04).

### Experimental setup

The experimental setup was similar to those described in *Tang & Fu (2019)*. Specifically, I used four rectangular tanks (length × width × height: 70 × 35 × 35 cm) as arenas to measure spontaneous swimming behavior in the different contexts, as in previous studies (*Liu & Fu, 2017*; Fig. 1). The arena was surrounded with an opaque canvas to eliminate visual stimuli from the observers during the experiments. The arena was separated into two subareas by an opaque plastic partition; the two subareas included an open area (length × width × height: 55 × 35 × 35 cm) and a shelter area (length × width × height: 15 × 35 × 35 cm), which provided a refuge with artificial plants. A small door (width × height: 10 × 10 cm) on the partition allowed fish to move freely from the shelter area to the open area in the boldness context, but it was closed in the exploration and activity (i.e., control) contexts. The water depth was maintained at 10 cm during the experiment. The behaviors of the test fish were recorded using a webcam (Logitech Pro 9000; Logitech Company, Suzhou, China) placed 1.5 m directly above the arena and connected to a remote monitor, and the experimental arena was illuminated by fluorescent lights.

### Experimental protocol

The recordings were conducted in all four arenas at the same time from 8:00 to 17:00 h. The fish were first recorded in the boldness context, then the exploration context, and finally the control context (see details below). The arenas were cleaned after the measurement of each individual.

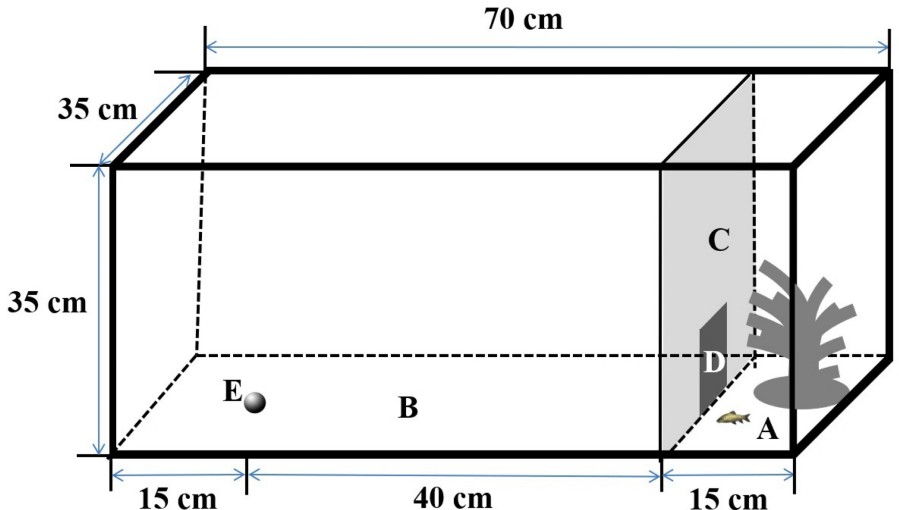

**Figure 1** **Design of the experimental aquarium used for the spontaneous swimming activity in boldness, exploration and activity contexts in the study.** (A) Shelter (length × width × height: 15 × 35 × 35 cm); (B) open area (length × width × height: 55 × 35 × 35 cm); (C) removable opaque PVC divider; (D) small door (10 × 10 cm); (E) novel object.

For the boldness context, a fish was transferred into the shelter area and acclimated for 30 min; then, the door was gently opened by a remotely operated line, and the movement of the fish (only in the open area) was recorded for 30 min (at 15 frames per second). Pale chub individuals showed large variation in latency to enter the open area (ranging from nearly zero to 23 min) and the percentage of time spent in the open area (18 to 97%). Thus, the total duration of moving in the open area varied greatly among individuals. To record movement in the exploration context, fish remaining in the shelter area were gently chased into the open area of the arena and acclimated for 10 min with the small door between the open and hidden areas closed. Then, a novel object (a black, round, plastic ball with a diameter of approximately 2 cm, *Liu & Fu, 2017*) was carefully placed on the arena bottom 40 cm from the small door (Fig. 1), and the movements of the fish were recorded by the webcam for 10 min. For the control context, the novel object was removed, and the fish was again acclimated for 10 min to eliminate the effect of the exploration context. Then, the movements of the fish were recorded for 10 min.

## Data calculation and analysis

Spontaneous movements were recorded for all 80 individuals in the three contexts. However, the number of replicates differed among the tests because some files could not be read by the software due to technical problems.

The videos were imported into an automated tracking program (EthoVision XT 9, Nodus, Netherlands) after converting the $x$ and $y$ coordinates from pixels to cm. The trajectories were smoothed using a weighted moving average with a window width of 0.5 s (*Miller & Gerlai, 2012*). Then, the median speed while swimming (swimming speed above 1.75 cm s$^{-1}$), percent time moving and median turning rate while swimming were

calculated for all three contexts. I eliminated movements with speeds less than 1.75 cm s$^{-1}$ to minimize potential system noise and small movements of the animal (body wobbles) according to previous studies (*Tang et al., 2017*; *Tang & Fu, 2019*).

The swimming speed ($v$, cm s$^{-1}$) was calculated as follows:

$$v(t) = \sqrt{(x(t) - x(t-1))^2 + (y(t) - y(t-1))^2}/d \qquad (1)$$

where $x(t)$ and $x(t-1)$ and $y(t)$ and $y(t-1)$ are the $x$ and $y$ coordinates, respectively, of the measured fish at time $t$ and the time of the previous frame $(t-1)$ and $d$ is the length of the time interval (i.e., 0.5 s, *Miller & Gerlai, 2012*). The percent time moving was calculated as the percentage of time when the swimming speed was above 1.75 cm s$^{-1}$. The turning rate ($\omega_t$, rad s$^{-1}$) of an individual was calculated using the absolute centroid change (see details in *Herbert-Read et al., 2013*). This measure can be used to quantify exploratory as opposed to goal-directed behavior in fish and is calculated as follows (*Ioannou, Singh & Couzin, 2015*; *Sumpter et al., 2018*):

$$\omega_t = \frac{\cos^{-1}(c_t \cdot c_{t+1})}{\Delta t}. \qquad (2)$$

In brief, $\theta_t$ is the orientation of the fish at time step $i$ measured relative to the positive $x$ axis of the coordinate system. Let $c_t = \cos \theta_t + \sin \theta_t$ be the unit vector pointing in the direction of $\theta$ at time step $i$. Then, the change in the orientation of the fish from time step $t$ to time step $t + 1$ is given by $\cos^{-1}(c_t \cdot c_{t+1})$. Again, I calculated the median turning speed of each individual only when the swimming speed was above 1.75 cm s$^{-1}$.

For the exploration trials, I also calculated two variables that are commonly used to test for differences in exploratory tendency in fish, i.e., the mean distance to the object and the latency to explore the novel object in the exploration context (*Adriaenssens & Johnsson, 2011*; *Mazué & Godin, 2015*). The distance (cm) was calculated as

$$D_i = \sqrt{(x_i - x_n)^2 + (y_i - y_n)^2} \qquad (3)$$

where $x_i$ and $y_i$ denote the coordinates of the fish and $x_n$ and $y_n$ denote the coordinates of the object. The mean value of all frames was used for analysis. The latency to explore the novel object was defined as the amount of time it took for the fish to first swim within 7 cm (approximately 1 body length) of the object. To assess whether the fish were responding to the novel object, I also measured mean distance and latency in relation to a virtual object in the same location in the control context. The calculation assumed that the coordinates of both the novel and virtual objects were located at the center of the object. Thus, I assumed that the 1 cm radius of the novel or virtual object would have little effect on measures of distance and latency to explore.

One-sample Kolmogorov–Smirnov tests indicated that median swimming speed, median turning rate and distance to the novel object were normally distributed, whereas percent time moving and latency to explore the novel (or virtual) object were not normally distributed. I used Pearson correlation to compare normally distributed measures and Spearman's rank correlation to compare measures that were not normally distributed. In the case of significant correlations, the relationships were also examined using linear

**Table 1  Correlations between measurements of activity in pale chub conducted in the control context and those conducted in either boldness or exploration contexts.** Correlations coefficients were calculated using Pearson ($R$) or Spearman's ($R_s$) correlation coefficients.

| | | Context | |
| | | Boldness | Exploration |
| --- | --- | --- | --- |
| Median swimming speed | | | |
| | $N$ | 74 | 78 |
| | $R$ | 0.644 | 0.656 |
| | $P$-value | <0.001 | <0.001 |
| Percent time moving | | | |
| | $N$ | 72 | 67 |
| | $R_s$ | 0.450 | 0.619 |
| | $P$-value | <0.001 | <0.001 |
| Median turning rate | | | |
| | $N$ | 74 | 78 |
| | $R$ | 0.660 | 0.664 |
| | $P$-value | <0.001 | <0.001 |

regressions. The effect of measurement context on median swimming speed and median turning rate was tested by a linear mixed model (LMM) using fish ID as a random factor. This was followed by a paired $t$-test to assess the difference in median swimming speed between any two contexts. The difference in percent time moving was assessed by a nonparametric Wilcoxon matched-pairs test. The distance from the real or virtual novel object was compared with a paired $t$-test, whereas the latency to explore the object was compared with a Wilcoxon matched-pairs test. The program SPSS 17 was used for data analysis. $P$-values < 0.05 were considered statistically significant, and all the data are presented as the mean $\pm$ S.E.

## RESULTS

### Effect of measurement context on spontaneous movement traits

The median swimming speed (Pearson correlation, $P < 0.001$), percent time moving (Spearman correlation, $P < 0.001$) and median turning rate (Pearson correlation, $P < 0.001$) measured in the control context were positively correlated with those measured in the other two contexts (Table 1; Fig. 2).

Measurement context had a significant effect on median swimming speed (LMM, $F_{2,150.29} = 70.637$, $P < 0.001$) but no effect on median turning rate (LMM, $F_{2,149.17} = 0.263$, $P = 0.796$) (Fig. 3). The median swimming speed of fish measured in the boldness context was significantly higher than that measured in both the exploration (paired $t$-test, $t_{73} = 9.723$, $P < 0.001$) and control (paired $t$-test, $t_{73} = 10.398$, $P < 0.001$) contexts. The percent time spent moving measured in the boldness context was significantly larger than that measured in the exploration context (Wilcoxon test, $z = -4.873$, $P < 0.001$), whereas the latter was significantly larger than that measured in the control context (Wilcoxon test, $z = -5.083$, $P < 0.001$).

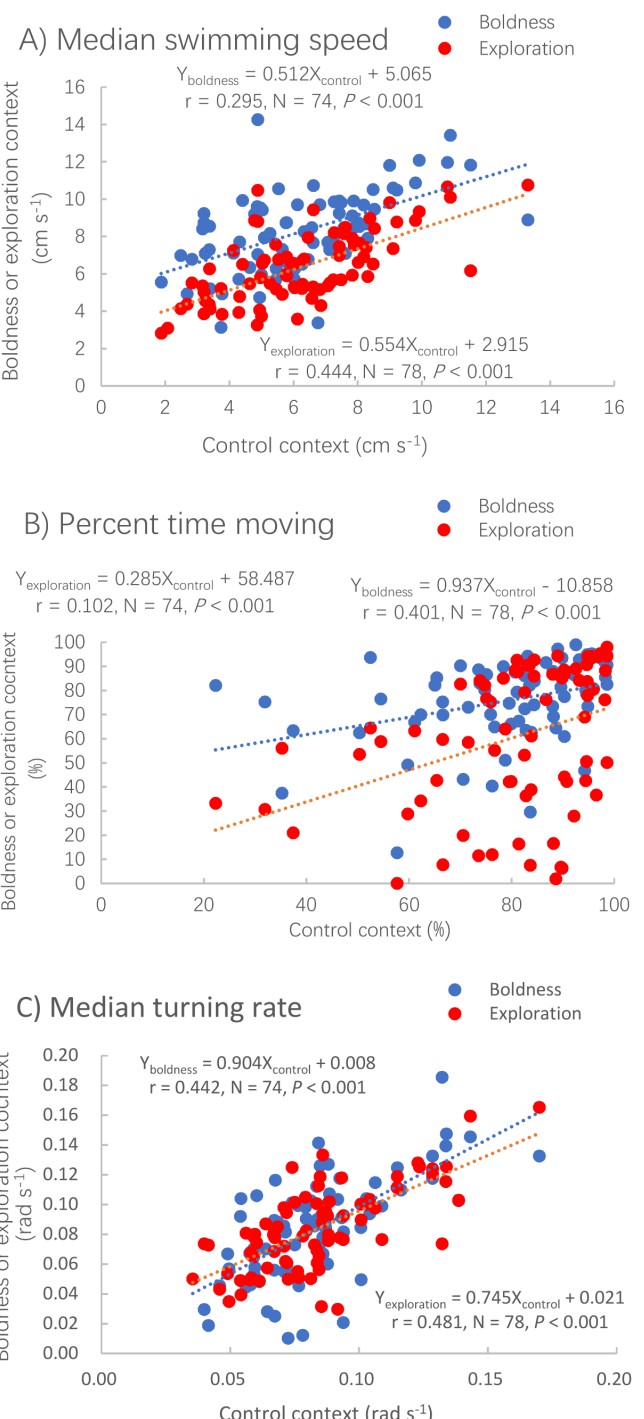

**Figure 2  Correlations between the control context and two other contexts, boldness (red dots) and exploration (blue dots) for three measures of activity.** Median swimming speed (A), percent time moving (B) and median turning rate (C) in pale chubs. Dotted lines represent the relationships between two different contexts using linear regressions.

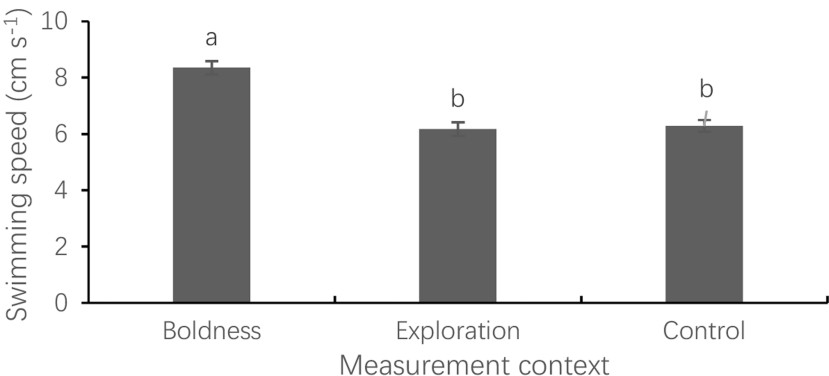

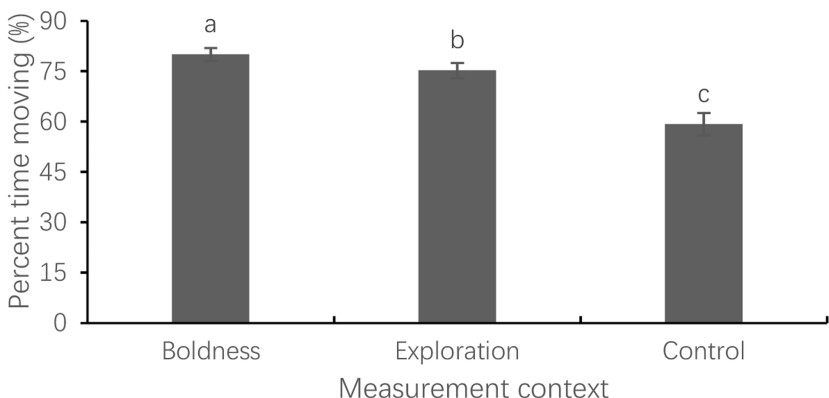

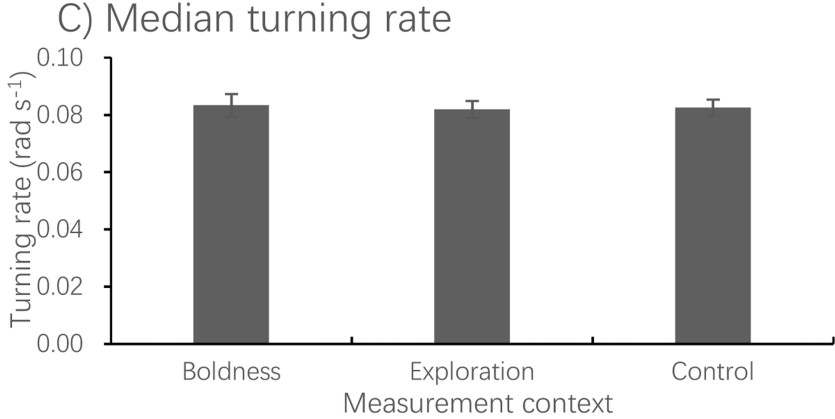

**Figure 3 Mean ± S.E. of three measures of spontaneous swimming activity.** Median swimming speed (A), percent time moving (B) and median turning rate (C) in pale chubs in the boldness ($N = 74$), exploration ($N = 78$) and control ($N = 79$), contexts. Bars with different letters are significantly different ($P < 0.05$).

### Difference in exploration variables between the exploration context and control context with a virtual object

The distance to the novel (or virtual) object (Pearson correlation, $R = 0.435$, $N = 79$, $P < 0.001$) and the latency to reach the novel (or virtual) object (Spearman correlation, $R = 0.329$, $N = 78$, $P < 0.001$) were positively correlated between contexts (Fig. 4).

The distance to the novel object in the exploration context showed no significant difference from that to the virtual object in the control context (paired $t$-test, $t_{78} = -0.838$, $P = 0.405$) (Fig. 5A). However, it took a much shorter amount of time for the fish to first reach the novel object in the exploration context than for the fish to reach the virtual object in the control context (Wilcoxon test, $z = 2.343$, $P < 0.001$) (Fig. 5B).

## DISCUSSION

### The correlation of spontaneous swimming traits between the control and other two contexts

The values of activity variables, such as the median swimming speed, median turning rate and percent time moving, measured in the control context were closely correlated with those measured in the boldness and exploration contexts. This agrees with the results of a previous study showing that variables such as median swimming speed and median turning rate were quite constant across measurements or contexts in mosquitofish (*Gambusia affinis*) (*Herbert-Read et al., 2013*) and in zebrafish (*Danio rerio*) (*Toms & Echevarria, 2014*). These results suggest that researchers can evaluate personality traits associated with activity using only boldness or exploration contexts and do not need to perform activity measurements separately. It is worth noting that the correlations may be related to factors other than individual differences because I measured the personality traits in only a single test. For example, it is possible that similarities between traits are due to factors such as the time of day, amount of time in the holding tank before testing, noise in the environment and stress of capture from the holding tank. However, it is unlikely that the correlation was due to such factors, as a previous study found that the median swimming speed and time spent moving under spontaneous swimming were constant and showed high repeatability throughout the day (from 8:00 to 18:00 h) and that almost all personality traits (seven of eight traits) showed high repeatability between two tests on two consecutive days (*Tang, 2019*). Nevertheless, more investigations (e.g., multiple measurements in different contexts but with different orders) need to be conducted before drawing a solid conclusion about the feasibility of using activity in boldness or exploration contexts as an adequate substitute for a separate activity test.

### Comparison of spontaneous swimming traits across contexts

Fish in the boldness context showed a 33% higher median swimming speed and 35% higher percent time moving than those in the control context. The faster speeds and more time spent moving of pale chubs in the boldness context could be because the fish could rely on returning to the shelter when threatened, whereas they reduced activity (speed and time spent moving) in the activity context because a refuge was not available. For example, a previous study in juvenile seabass (*Dicentrarchus labrax*) showed that activity is reduced

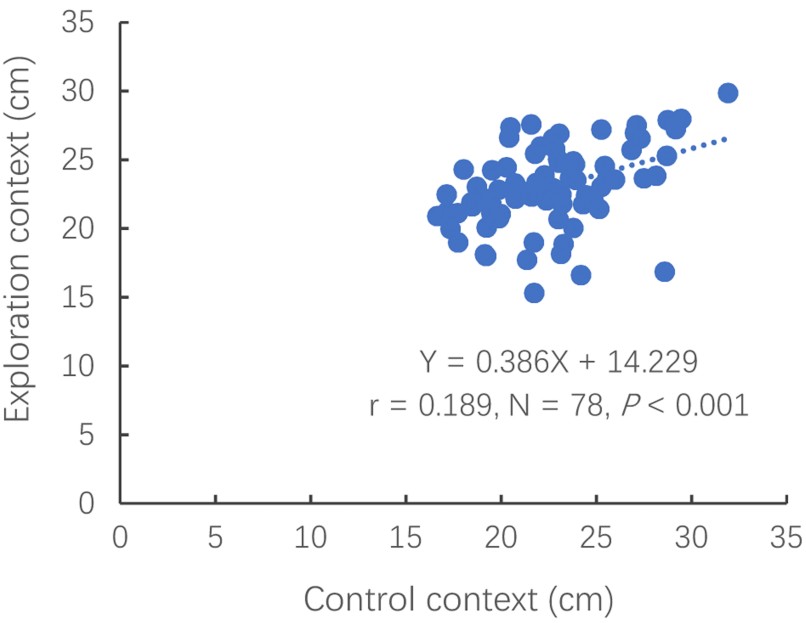

### A) Mean distance to object

$$Y = 0.386X + 14.229$$
$$r = 0.189, N = 78, P < 0.001$$

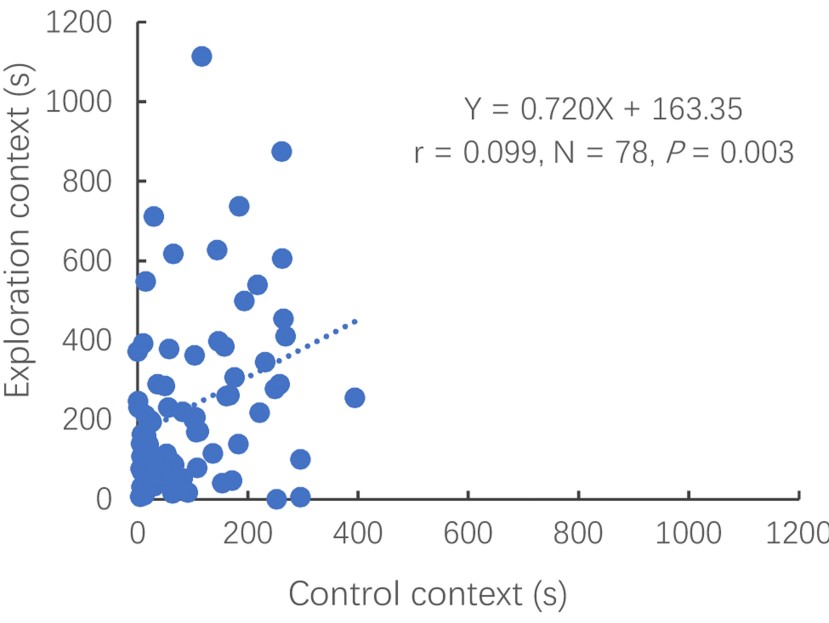

### B) Latency to exploration

$$Y = 0.720X + 163.35$$
$$r = 0.099, N = 78, P = 0.003$$

**Figure 4  Correlations between the exploration and control contexts for measures of distance to the novel object (A) and latency to the object (B).** Dotted lines represent the relationships between two different contexts using linear regressions.

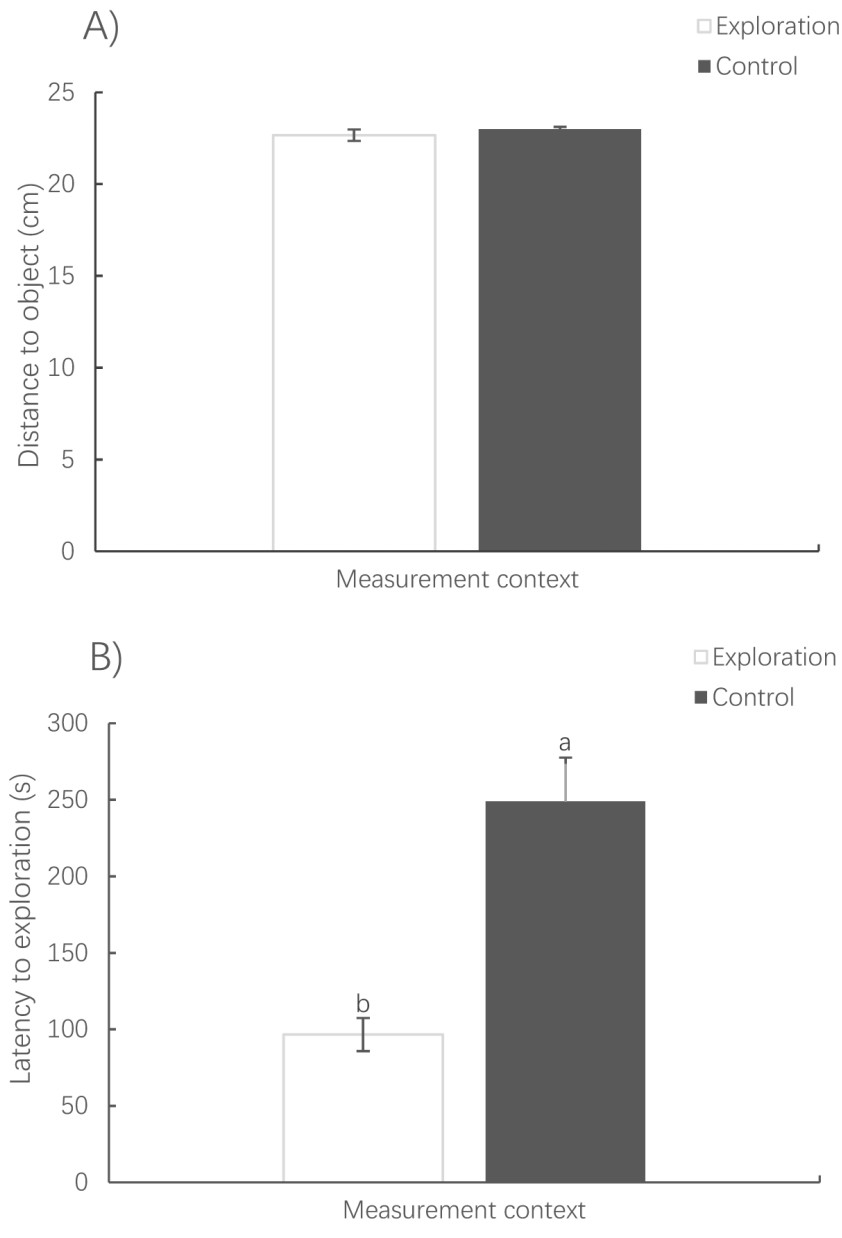

**Figure 5 Mean distance (A) and latency to exploration (B) under the exploration context and the virtual object under the activity context.** (Mean ± S.E., $N = 78$). Bars with different letters are significantly different ($P < 0.05$).

when perceived threat increases (*Herbert-Read et al., 2017*). However, the difference in swimming activity between boldness and control contexts might also be because some higher movement rates result from fish fleeing quickly back to the shelter after entering the arena. Finally, the differences between the boldness and control contexts might also be a result of test order or time in the arena because the three contexts were measured in the same order for all fish. For example, although the pale chubs were acclimated for

30 min (the acclimation period was long enough according to our previous observation) before measurement in the boldness context, one might argue that the fish were more stressed from being captured and hence moved and explored their environment more in the boldness context but then either tired or calmed down in the exploration and control contexts. This might be an alternative explanation for the reduction in swimming speed or percent time moving from the boldness to exploration or activity context. Future studies should consider the effect of order in the arena on personality measurements.

## The relationships of exploration variables between exploration and control contexts

All variables measured in the exploration context were almost the same as those in the control context. This suggests that the presence of so-called novel objects might not have been a stimulus for exploration in pale chubs in the present study. This was not due to the size and shape of the arena (70 × 35 × 35 cm) or novel object (1 cm radius) in the present study, as they were similar to those in previous studies. For example, a previous study in blue gourami (*Trichogaster trichopterus*) used a 40 × 60 × 40 cm arena (*Bisazza, Lippolis & Vallortigara, 2001*), and a 90 × 30 × 45 cm arena and 1.3-cm-diameter novel object were used for the guppy (*Poecilia reticulata*) (*De Serrano, Fong & Rodd, 2016*). A 50 × 20 × 36 cm aquarium and 1.8-cm-diameter novel object were used for zebrafish (*Lucon-Xiccato & Dadda, 2014*), and identical conditions were adopted in crucian carp (*Liu & Fu, 2017*). Thus, the fact that previous studies as well as the present study found a positive relationship between activity variables such as speed (or percent time moving) and exploration variables (distance and latency to explore the novel object) might have occurred because more active fish spend more time swimming at higher speeds and hence show a shorter distance to the object and exploration latency than do less active individuals (*Liu & Fu, 2017*; *Tang & Fu, 2019*). This is reinforced by the results indicating that the fish showed almost identical values in terms of the mean distance to novel objects in the same location when measured in the control context (i.e., a virtual object). However, this might have occurred because the pale chubs in the present study did not need to change their swimming behavior or position in the arena to explore. Furthermore, the handling stress during experimental measurement might also have resulted in a lack of change in swimming pattern in the fish when subjected to different test contexts. The situation could have been reinforced, as pale chubs were measured individually in the present study, whereas this fish species often forms shoals in the field. Recent studies in cyprinid fish species such as qingbo (*Spinibarbus sinensis*) found that the responses of spontaneous activity patterns to threat varied profoundly when tested individually compared to those tested in groups (*Wang, Fu & Fu, 2019*; *Xu, Fu & Fu, 2019*).

Interestingly, the time it took for the fish to first reach the object in the exploration context was much shorter than that in the control context; i.e., the fish encountered the object earlier in the exploration context. This suggests that pale chubs not only showed no neophobic tendency but also exhibited exploratory behavior towards the 'novel' object and that latency can be used as an exploration indicator in the pale chub under the conditions used in the present study (*Galhardo, Vitorino & Oliveira, 2012*). This might be because the

pale chub perceived the 'novel' object as something that was potentially rewarding and worthy of exploration, such as food. The lack of a difference in the spontaneous activity pattern of the pale chub between the exploration and control contexts might have occurred because the novel object was too simple and the pale chub performed only superficial inspection or contact with the object and then exhibited a spontaneous swimming pattern similar to that observed in the control context. Nevertheless, this study suggests that fish behaviorists should be cautious regarding explanations pertaining to exploration (or neophobia) indicators previously used in other personality studies, such as the distance to an object (*Wang et al., 2019*; *Wang, 2019*). Researchers at least need to test the reliability of such measurements, for example, by comparing spontaneous movement trajectories in fish between exploration and control contexts, as in the present study. The testing of other variables under different experimental setups, such as the number of times that individual guppies traveled between containers connected by pipes (*Brown & Irving, 2014*), might be more appropriate for the measurement of exploration tendency.

## CONCLUSIONS

In conclusion, the pale chubs showed strong positive correlations in median swimming speed, median turning rate and percent time moving between the control and other two contexts, which suggests that fish behaviorists can use such traits from either boldness or exploration measurement tests rather than having to conduct additional tests. Further analysis revealed that the latency to explore an object might be a reliable indicator of exploration, whereas a traditionally used variable such as distance to a novel object might not be an appropriate indicator, at least for the pale chub under the conditions in the present study.

## ACKNOWLEDGEMENTS

We would like to thank Dr. Donald Kramer, Dr. Christos Ioannou and one anonymous reviewer for insightful comments and constructive suggestions for statistical analysis and writing that greatly improved the manuscript.

### Funding
This work was supported by the National Natural Science Foundation of China (No. 31670418). The funders had no role in study design, data collection and analysis, decision to publish, or preparation of the manuscript.

### Grant Disclosures
The following grant information was disclosed by the author:
National Natural Science Foundation of China: 31670418.

### Competing Interests
The author declares there are no competing interests.

## Author Contributions

- Shi-Jian Fu conceived and designed the experiments, performed the experiments, analyzed the data, prepared figures and/or tables, authored or reviewed drafts of the paper, and approved the final draft.

## Animal Ethics

The following information was supplied relating to ethical approvals (i.e., approving body and any reference numbers):

This study was approved by the Animal Care and Use Committee of the Key Laboratory of Animal Biology of Chongqing (permit number Zhao-20161122-04).

## Data Availability

The raw measurements are available as Supplemental Files.

## Supplemental Information

Supplemental information for this article can be found online at http://dx.doi.org/10.7717/peerj.8736#supplemental-information.

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
