# Peer review of "The effect of personality measurement conditions on spontaneous swimming behavior in the pale chub Zacco platypus (Cyprinidae)"

_PeerJ, doi:10.7717/peerj.8736_

## Round 0.1 · original submission · Major Revisions

Overview
This study examined how three tests for personality traits (exploration tendency: novel object present, boldness: shelter present, and control: neither novel object nor shelter present) affected four measures of spontaneous swimming behavior (percent time moving, swimming speed, turning rate, and distance to the center of the arena) in a cyprinid fish, the pale chub. Swimming behavior did not vary between exploration and control treatments, and mean and minimum distance from a novel object did not differ from the same measures in relation to a virtual object in the control treatment. However, latency to approach the novel object was less than the time to approach the position of the virtual object. The author suggests that latency to approach a novel would be a better measure of exploration than average position in relation to a novel object because position could be an indirect result of activity rather than exploratory tendency. During the boldness test fish averaged 46% of their time in shelter and showed higher percent time moving, swimming speed and turning rate when out of shelter but no difference in distance to center. The author suggests that pale chub prefer shelter and that latency to enter the open area or time spent in the open are appropriate measures of boldness. All four measures were highly repeatable within individuals across the three tests. There were strong correlations among individuals in all three tests for all four measures except that distance to center was not significantly correlated between the boldness and control tests. The author suggests that the consistency of activity measures during novel object and shelter contexts imply that future tests of fish personality may not need a separate control test for activity

Both reviewers considered that the study was basically sound but that substantial changes were needed to clarify its message, to relate the study better to previous literature, to provide missing information about the methods and results, and to improve the statistical analysis. I found that the writing style was generally good with few errors and that the number of test subjects was excellent but that the organization of the manuscript needs to be strengthened and many missing details of the study and results need to be added and that some additional analyses could greatly strengthen the conclusions. My comments below build on those of the reviewers, including many of the points that they raised, but also additional topics that they did not mention. In addition, I have provided a pdf with yellow highlights to indicate problems and inserted comments to suggest alternative wording.

Major Concerns
1. Organization
1. The list of objectives needs to be complete and logically related to the methods. In the present manuscript, two goals are provided but they seem to be two potential benefits of a single objective: to determine whether swimming behavior is a repeatable trait within individuals across different personality tests. The three analyses, i) correlation among individuals between tests, ii) differences among tests in swimming variables, and iii) controls for measures of approach to a novel object, need to be related to specific study goals. What you need is a paragraph starting on L63 of the present Introduction in which you relate your question about the potentially artifactual correlation among personality traits to any previous literature on this topic and then to explain how your study addresses the question. In other words, what precisely is the knowledge gap you wish to address? You briefly state a hypothesis but do not explain the logic of that hypothesis or how it relates to your question. What results would support the artifact hypothesis and what results would support the non-artifact but correlated personality variables hypothesis?
2. Different objectives need to be addressed in a logical and consistent order. The goals should be presented in an order that is logical and clear and then the rest of the manuscript (including Abstract, summary of previous research in the Introduction, statistical methods, Results including tables and figures, and Discussion) should follow the same order.
3. Terminology for the testing treatments needs to be more consistent and concise. The manuscript is inconsistent about terms for the different the test contexts: novel object vs. exploration, shelter vs. boldness, activity vs. control). There is substantial redundancy in the manuscript as a result of repeating the personality variable and tank setup. I suggest calling the tests exploration, boldness and control, carefully describing the test in the Methods, and then using these terms consistently in the results, figures and tables, and discussion. (Of course, in the Abstract you will need to briefly indicate how the personality and test situation are linked.) Reviewer 1 suggests ‘activity’ rather than control as the preferred term. Because you measured activity in all treatments, this might be confusing but you can decide which is most appropriate based on prior use of the terms in the literature. You can refer to the three treatments collectively as ‘treatments’, ‘tests’, or ‘contexts’, but be consistent so the reader understands what you are referring to.
4. It is crucial that the meaning of ‘repeatable’ be consistent with the literature and clearly defined for the reader. This should occur in the Introduction and be unambiguous even in the Abstract. In the Discussion, it sometimes appears that you are using repeatable to refer to similar means and other times to positive correlations. However, correlations can be positive when means are different and means can be similar even when correlations are not significant. To my understanding, only correlations of individual responses between tests provide evidence of personality.
5. Reviewer 1 raises serious questions about whether exploration is the correct term for the test with the novel object. As he advises, please check the literature and change the term or provide a justification using established literature for this term.
6. When considering the three treatments, use a consistent order in the Abstract, Methods, Results including tables and figures, and Discussion.

2. Methods
Many important components of the Methods are missing, some of which were noted by the reviewers. Although excessive detail should be avoided, a reader should have enough information to repeat the study if he or she chooses to do so.
1. Origin of study species. Please specify approximately where the fish were captured, for example, by stating a particular stream or watershed or indicating the vicinity of a particular city or town. Provide latitude and longitude to assistant international readers identify the location.
2. Please provide a brief background on the ecology and behavior of the study species. For example, is it a territorial or schooling species, what is its diet, is it found mainly in running or still water bodies, does it inhabit highly structured habitat or open water, does it use structured habitat such as weed beds or flee to open water when threatened?
3. Did you use only one test aquarium used for all 80 x 3 trials as implied or were there multiple similar aquaria?
4. Be more specific about what you mean by a hidden area with shelter. What was the shelter? In what way was it hidden?
5. The methods should state the presence of the door in the partition and the mechanism by which it was opened.
6. What was the novel object (e.g., size, shape, color) and why was it selected? This becomes important for the comparison between the real and virtual novel object.
7. When was the novel object placed in the tank, before the fish was introduced or after the 10-min acclimation period?
8. Where was the novel object placed in the aquarium?
9. Was the webcam located directly above the tank?
10. In the boldness treatment, what was the procedure if the fish returned to the shelter area before 10 minutes of data had been collected? Did you continue to record until you had 10 minutes of data in the open (which readers might assume because you stated that you collected 10 minutes of data in the open) or did you record for 10 minutes only and only measure activity in the arena? This might be important to interpreting the data because the sample size for the swimming variables will differ, so it must be very clear.
11. For the calculation of movement variables, what was the time interval between frames? What was the total number of measures per individual in each trial?
12. Was the swimming speed calculation overall speed or speed while moving, i.e. was it based on all intervals or only the intervals which were above the threshold for percent time moving (1.75 cm/s)? If it is based on all intervals, then swimming speed would not be mathematically independent of percent time moving.
13. In the section on measurements, specify what measure you used in your analysis. For example, the introduction implied that you used the median swimming speed, not the mean, and that should be stated in the Methods. Did you also use the median for turning and distance?
14. Provide the units for distance from center and distance from the novel object.
15. Provide more details concerning the calculation of turning rate, including the units. The reader should not have to read a different article to understand your measure.
16. Is the measure of turning really head-turning or is it the turning of the whole body? It seems that you could use the simpler term ‘turning rate’ throughout the manuscript unless there is a good reason to clearly differentiate head-turning from other measures of turning.
17. For comparing the position of the fish from the real and virtual novel object, more details are needed. The calculation assumes that the novel object was located at a specific point (presumably the center of the object) but it presumably had some dimensions. How was this taken into account in comparing the real and virtual tests?
18. Please justify the 7 cm criterion for contact with the novel object. A circle with a 7 cm radius includes nearly half of the width of the aquarium. Is this a valid measure? It seems that you might have used a test designed for a larger circular arena in a relatively small aquarium. If so, this needs to be considered in your discussion.
19. The Methods section is missing a section on the experimental protocol. This should have all information for another researcher to repeat the study, including information on the sequence and timing of treatments (amount of time between successive treatments), time of day in relation to the fish’s photoperiod, where individual fish were held between treatments, any cleaning of the tank or replacement of water between tests, and, if different tanks were used for different fish or treatments, whether the fish experienced the same or different individual aquaria in the different trials.
20. If treatment order was randomized, you should indicate briefly the randomization procedure, and you can test statistically for an effect of treatment order. If treatment order was systematically varied so that one third of the fish, for example, experienced each treatment first, the protocol should be stated. Using the same order for all treatments is an error in experimental design because treatment is confounded with experience in the test situation, and this could influence exploration and swimming. The error would not be serious enough to reject the manuscript, but the protocol must be clearly stated and its implications for interpreting the results included in the Discussion.

3. Statistical Analysis
The reviewers have identified several concerns regarding the statistical analysis and presentation. Although I am not particularly expert in statistics, I have some additional questions for which I advise you to seek statistical advice.
1. For the ANOVA, each individual was tested once in each of the three treatments. However, you used a one-way ANOVA. This assumes that the data are independent, but they are not. Accounting for individuals would require a two-way ANOVA with individual identity as a factor. Post-hoc tests also must take individuals into account with a paired analysis. However, a linear mixed-effects model (LMM) where individual identity is a random variable and the treatment is the fixed effect would be a more robust method than a two-way ANOVA. It is possible that there could be a treatment effect when the analysis accounts for individuals. Please consult a biostatistician regarding the best approach. Whichever method you choose, you will gain additional insight into individual repeatability by the effect of individual identification. When presenting the results of an ANOVA, you should provide F and df as well as the p-value.
2. The ICC is only briefly mentioned in the methods. We need more information on how this assesses repeatability. Reviewer 1 indicates that your method of assessing repeatability is outdated and that more powerful tests exist. He doesn’t mention what those tests are, so you should check the recent literature in major journals to see what stronger approaches the reviewer is referring to. Then consult a statistician regarding the most appropriate approach.
3. For the correlations, I am not as concerned as the reviewer that some points may be hidden in the graphs. This can be clarified by specifying the sample sizes in the caption. However, there are other issues:
o Why did you not compare the measures for the boldness and exploration contexts? Consider adding this comparison.
o You show trend lines, which are useful, but you do not indicate how these lines were calculated. Are they regressions? If so, what type? You should provide the equations for regression lines.
o In addition, the lines indicate slopes that are less than a 1:1 line. (I drew a 1:1 line on the graphs to see this more clearly.) This implies that as individual measures increase in the control treatment, they increase more slowly in the exploration or boldness treatments. The graphs suggest that fish that had lower values on the control test tended to have relatively higher values in the other tests than fish that had higher values on the control tests. This could be an effect of chance, but it might be an interesting biological pattern to explore if the difference from a 1:1 relationship is significant. Please consult a statistician to see if there is more information here that could add insights to your conclusions.
4. Please check that the data meet the assumptions of the tests, make appropriate transformations, if necessary, and inform the reader. Often, percent and proportions are not normally distributed. Fig. 3B suggests that this might be the case with your data as well (lots of points near the 100% line).
5. When reporting the results, please provide the test statistic (e.g. F-value), sample sizes or degrees of freedom as appropriate, as well as the p-value.

4. Results
It is not clear why you omitted some comparisons and results.
1. The absence of a comparison between exploration and boldness in the correlation analysis was mentioned above. You also did not mention the post-hoc comparisons between these treatments for the ANOVA.
2. The reviewer notes that you could and should check the repeatability of your measures of exploration of the novel object by checking the correlation between the measures in real and virtual novel object tests.
3. You might gain additional insights by checking correlations between the activity variables in the three treatments and the three measures of response to the novel object.
4. As noted by reviewers, time to leave the refuge is a relevant variable in your discussion. It is not clear why you did not present this information as you apparently recorded it (L218). Comparing this measure to the other swimming variables in the boldness test could be of interest.
5. You should test whether time in the shelter differed significantly from the expectation of 21% instead of just presenting the average.

5. Discussion
A. Exploration vs. Control
1. It is important to consider all possible explanations for this result. One possibility is that the object is not a sufficient stimulus for exploration. (Note that this wording is less anthropomorphic and clearer than suggesting that the fish ‘did not care’.) Alternatively, the stimulus may have been adequate to stimulate exploration, but the fish did not need to change their swimming behavior or position in the aquarium in order to explore. For example, swimming behavior in the aquarium might already be exploratory because the aquarium is a novel environment for the fish, so adding another element of novelty may not change the situation from the fish’s perspective. This might be a particularly important issue in a relatively small aquarium where there may not be opportunity or need for changes in movement in the presence of a novel object. (Consider the implications of the size of your test aquarium in relation to the size of the fish compared to previous studies.) Consider whether the novel object could be perceived by the fish as habitat structure (cover) that might be attractive in the open area. Your experience with the experiment and your knowledge of the literature might lead you to discover additional explanations. Please provide a more rigorous and thorough discussion of how to interpret the lack of change in swimming but a change in latency to approach the object.
2. Is it possible that pale chub juveniles typically join schools or shoals so that the fish are responding to the stress of being separated from their conspecifics? Stress could lead to a pattern of swimming that is little affected by other test conditions.
3. The conclusion that measures of exploration in other studies may be artifacts of general swimming activity would be supported by an analysis of the correlation between the swimming variables and the exploration variables.
4. You should consider carefully whether the choice of novel object could have affected the limited response of the fish and raise this issue appropriately in the Discussion. Before drawing general conclusions about the inadequacy of the measures of exploration used in previous literature, you need to consider whether experimental variables such as the type of object and size of the aquarium may have resulted in the limited effect found in your study.
5. It is critically important that research results be related to previous research on the topic so that the contribution of the study is clear, whether it is highly original or confirmatory of previous findings. This section needs a paragraph relating your findings to previous research. You should relate your findings to those of any other researchers who compared locomotion or activity in different personality test regimes. If this has not been done before on fish, you should also investigate whether it has been done on other taxa. Have previous researchers compared measures of exploration with and without a novel object? Did they find similar or different results. Your understanding of the literature should be sufficient to allow you to specify the originality of your contribution to the scientific literature.

B. Boldness vs. control
1. The higher values of activity in the shelter experiment seem quite small. Although statistically significant, you should discuss whether they are they biologically significant.
2. Distance to center did not show significant differences in either comparison, and Fig. 3 shows that there was relatively little variation among individuals. Is it possible that the size and shape of the aquarium greatly constrained this measure so that the values have little meaning as compared to a larger arena where a small distance from center involves moving farther from the walls?
3. The conclusion that the fish prefer shelter requires a statistical analysis.
4. The conclusion that fish rest in the shelter does not seem to have support from any measures of activity in shelter. It is possible that less bold fish hesitate to emerge from the shelter but they are not resting there.
5. You present only one interpretation of the higher activity levels of fish in the boldness test, that they search for food or predators there. Is there any supplementary evidence to support this interpretation? Are there any alternative explanations? Is it possible that some of the higher movement rates result from fish fleeing quickly back to shelter after entering the arena?
6. The conclusion that time in shelter is a useful measure seems to be based only on the substantial variation in this measure and the difference of the mean from the expectation. This is very limited evidence because it provides no indication that this is repeatable within individuals. I suggest testing whether fish that entered the arena earlier also spent more time there after entering and whether any of the swimming variables were correlated with any of the boldness variables?
7. As with the exploration test, this section must be integrated with previous research on this topic.

C. Repeatability
1. L228-232. These sentences confound the mean difference results with the correlation results. I don’t think that these both relate to repeatability as used in the context of fish personality. The role and interpretation of these two aspects of the analysi should be clearer in the Introduction, Methods and Discussion.
2. What do the correlations mean for evidence of personality of pale chub? How do they relate to previous literature looking at repeatability within the same type of test as compared to using different types of tests? As with previous sections, the Discussion needs a strong connection to the previous literature.

D. Conclusions
1. You did not specifically discuss in how repeatability of all variables raises questions about the validity of the exploration measures so it is not appropriate to simply state it here. Please justify in the Discussion. This argument seemed to be based on the similarity between the mean measures in relation to the real and virtual novel object rather than the correlations so is confusing.
2. PeerJ instructions to authors indicates that Conclusions is a separate section, not a sub-heading of Discussion. It should identify unresolved questions / gaps / future directions. The present conclusion mostly repeats the Discussion and Abstract and is highly redundant.

Other concerns and suggestions

Title: The first part of the title is too broad. Your review does not show that there is a doubt whether personality variables are reliable in fish. It is not clear what ‘reliable’ means in this context. Your study only answers a question about repeatability in one species, not in fish in general. In the second part, ‘spontaneous behavior’ is too general and vague. I suggest ‘The effect of personality test conditions on spontaneous swimming behavior in the pale chub Zacco platypus (Cyprinidae)’

Abstract
L23. It seems unnecessary to abbreviate percent time moving which simply adds another acronym for the reader to learn to understand the paper.

Keywords: consider terms not used in your title or abstract that other researchers might use to find your study, for example, ‘behavioral syndrome’ and ‘temperament’. The family Cyprinidae will be important if you do not mention it in the title or abstract.

Introduction
L48. Do not abbreviate percent time moving or total distance moved.
L53. I don’t think that ‘observation’ is the right word to describe this behavior. Observation can take place from a distance.
L53. Consider starting a new paragraph when you switch from describing individual tests to relationship among traits.
L54. Is ‘explorative’ the word normally used in the literature? I would have expected ‘exploratory’.
L58-60. Do you mean that the observed relationships among traits might be an artifact of the measurement protocols? This might be a stronger way to express the concept.
L63. Are there any literature references in which this suggestion has been previously made? Are there examples in which the literature in which personality correlations could be the result of such an artifact? Is there any literature which contradicts this idea by presenting data in which the correlations could not be explained by the artifact?
L63. Start a new paragraph here explaining more completely how your study relates to the question of whether personality correlations could be due to an artifact.
L70-72. I do not see how your study provides insight into the possibility of combining exploration and boldness tests. It seems only related to combining activity and each of these tests separately. You have not addressed the possible problems involved in combining exploration and boldness in a single test.
L77. E.g. means ‘for example’. This is a complete list, so ‘e.g.’ is not needed.

Methods
L80. Indicate what the variation means by writing ‘mean +/1 S.D.’ the first time you use it. Standard deviation is most appropriate if the data are normally distributed but it would be useful to add the range (minimum and maximum values).
L91, 106. Acclimatization is usually used to refer to the response to natural changes in the environment. You should use the term ‘acclimation’ for adjustment to laboratory conditions as you did later in the manuscript (L117, 122).
L125. This should be the start of the Data Analysis sub-section.
L139, 153. A parenthesis is missing in the equation
L146. You do not need this sub-heading.
L147. It is preferable not to present results in the methods. I have suggested an alternative wording that avoids this.
L150. I assume that you don’t mean to refer to personality tests in general but specifically to exploratory tendency. Provide references for the use of these measures.
L151. ‘First time exploring’ is not clear. I suggest ‘latency to explore’.
L157-159. The word ‘fake’ is too colloquial for a scientific paper. I suggest using the term ‘virtual’ throughout and have suggested alternative wording to explain what you did.
L168. This is incorrect because you did not compare between boldness and exploration.

Results
L172. I presume that these are ICC result but that is not clear because ICC was mentioned last in the statistical section. Please specify. Also, provide the sample sizes. I don’t know if ICC can use all the data or only data from individuals for which all three tests were successful.
L174. You need to show the complete results for the ANOVA tests, including F, d.f. and in either the text, a table or the figure (but in only one of these).
L217-219. Data on the time in shelter and latency to enter the arena (including mean for the latter measure) should be presented in the Results, not Discussion.

Discussion
L230. Indicate what organism and context were studied and be clear whether repeatability refers to correlations or average similarity in this study.
L234. If this is simply unpublished research, it would probably be better not to refer to it. If the research is being prepared for publication or has been submitted and you feel it is important, then indicate its status (‘in preparation’, ‘submitted’, or ‘in press’) rather than simply referring to it as ‘unpublished’.

Tables and figures
Table 1. The heading is incomplete. Specify the type of correlation, the measurements and the other two contexts. Don’t abbreviate percent time moving. P-value has a hyphen.
Fig. 1. Add dimensions of the door to the caption.
Fig. 2. PeerJ instructions indicate that authors should use capital letters to identify panels in figures. There are several grammatical errors in the caption; I have suggested an alternative wording on the pdf.
Fig. 3. Capital letters for panels. It is sufficient to have the key only on panel A. I have suggested an alternative wording which I think is clearer. You need to add information regarding what the dotted lines represent and provide the equations for the lines. Alternatively, you could add this information to Table 1.
Fig. 4. Please revise the caption taking into account the suggestions above for Fig. 2 and 3.

Supplementary data.
For the main dataset, include the individual id so a reader can be sure that all data on a row come from the same individual. Also, add a key to provide units for each measure and to define any abbreviations in column heads. The data file for Fig. 4 also needs the units. ‘Time’ is misspelled in the header.

·

Basic reporting

The study itself seems robust, although some essential details are missing from the methods that need to be included to make this judgement. There is much room for improvement in the analysis (including further data that was collected but not analysed), interpretation of results and presentation. I think the importance of the findings may be obscured by the current presentation.

There are a number of improvements to be made in the presentation of the text, including grammar and choice of words/terminology. I have made various comments below but these are by no means exhaustive. Figure legends are multiple short paragraphs and different fonts.

L 83, “the local fisherymen” should be “local fishermen”. L 85, “Two-tenths” should be “One-fifth” or “20%”. What does “properly” refer to in “properly placed”? Delete “i.e.” in “i.e., the pale chub”. Generally, “i.e.” is overused throughout.

The abstract is rather long and detailed, and should be made clearer and easier to read, and possibly made shorter.

The Introduction is too short and mostly one very large paragraph. The Introduction should be longer and broken up into multiple paragraphs to aid readability.

The Discussion should include a paragraph on why the measured variables are important, i.e. what consequences do they have for fitness? For example, encountering food:
Search rate, attack probability, and the relationship between prey density and prey encounter rate
CC Ioannou, GD Ruxton, J Krause, Behavioral Ecology 19 (4), 842-846

Experimental design

This was generally good but some details are missing to make a full assessment.

Validity of the findings

See specific comments.

Additional comments

L 20: What is meant by “spontaneous” in “spontaneous movement traits” and elsewhere? It is used frequently without its meaning being clear.

L 28-30: This is unclearly written.

L 36: It is surprising to see the repeatability of the behaviours being presented at the end of the abstract, like an afterthought, when the title focuses on personality.

L 37: Rather than “the personality of pale chubs is quite conservative”, is “the personality of pale chubs is conserved” meant?

L 42: Cite reviews / meta analyses here for this general point, such as
Bell AM, Hankison SJ, Laskowski KL. 2009 The repeatability of behaviour: a meta-analysis. Anim. Behav. 77, 771–783.
rather than primary literature using fish; there is no reason the literature cited should be so fish orientated.

L 50: “inspection” is referred to here, but inspection of what? I think the general consensus is exploration is movement in an unfamiliar area, but behaviour toward a novel object measures neophobia. The literature should be checked, and the choice of terminology justified in the manuscript with the supporting references (ideally highly cited reviews on the subject).

L 56: Is this not referred to as a behavioural syndrome? This term is not in the manuscript except the references, which is surprising.

L 64: It is not clear what “or relevant” means here.

L 68-69: How are these different in practice?

L 122: Measures such as the time taken to first leave the refuge, or to reach a certain point in the arena, are used in such assays of boldness, rather than the behaviour of the animal when outside the refuge. See, for example:
Bevan PA, Gosetto I, Jenkins ER, Barnes I, Ioannou CC. 2018 Regulation between personality traits: individual social tendencies modulate whether boldness and leadership are correlated. Proc. R. Soc. B Biol. Sci. 285. (doi:10.1098/rspb.2018.0829)
This should be measured and its relationship with the trajectory data tested for.

L 133: It is essential to include here what the time interval was between fish being tested. Also, I don’t think it is stated whether tests were done in a randomised order for each fish? Does testing order have any effect?

L 129: It’s not clear how relevant “distance to the center of the open area” is when it becomes apparent that the test tank is long and thin; even in the centre of the arena, the fish will still be close to a wall. This variable would be more appropriate in a circular or square arena, where the centre is far from the nearest wall.

L 136: There’s a reference here for the choice of this threshold, but it needs more justification.

L 143: There needs to be more justification for the choice of term “head-turning rate”. Presumably the tracking software tracks the centre of mass of the fish, so this variable is absolute turn angle of the trajectory. Due to the morphology of fish where the body is not rigid, this is not necessarily the same as the head-turning rate.

L 147: “almost identical” is a poor choice of words in “were almost identical between the activity and exploration contexts”. Almost identical in what sense? Same mean and variance? Or strongly correlated? Or something else? This also sounds like it should be in the results, and it doesn’t need to be here.

L 160: How have the assumptions of the tests been tested for, and are the assumptions met? This is essential to include.

L 166: The repeatability of the response to the novel/fake object isn’t tested for? Also, the method of testing for repeatability seems rather outdated and there are more sophisticated and powerful tests that allow for control over covariates such as body size.

L 172 onward: Why is repeatability presented first, then average differences, then repeatability again?

L 176: Rather than ‘control condition’, it would be better to refer to ‘activity context’ (or similar) to match the figures better. Consistency helps the reader.

L 185: Why isn’t repeatability for these variables explored too?

L 187: This seems to directly contradict the statement in the abstract that “pale chubs might ignore the existence of a so-called ‘novel’ object.”.

L 193: “might not have cared” is anthropomorphic.

L 195: Doesn’t this contradict the result in Fig 4c?

L 196: It is written as if the results presented are different to those found elsewhere, but it’s not made clear this is the case.

L 210: This statement needs more justification.

L 217: Since this data was collected, it should be analysed (see comment above). It would be of interest to see how it correlates with data from the fish’s trajectories when outside the refuge.

L 228: Again, “All variables in the present study were quite similar across the different contexts.” is too vague.

L 232: I think this should be deleted, as unpublished data is being referred to that has not been peer reviewed.

L 237-239: This needs more explaining, it seems contradictory.

Figure 3: It looks like a lot of data points are obscured by the opaque filled circles.

Christos Ioannou
University of Bristol

Reviewer 2 ·

Basic reporting

Overall, I commend the author on the quality and accuracy of the English writing. That being said, there are a few lines that require editing, below are a few examples:

-line 60- ‘trauts’ should be 'traits'
-lines 69-70- add ‘be able to’ to “researchers may ______ acquire data…”
add ‘if correlated’ to “For example, ________, one can measure”

Also, there is some inconsistency in the format of in-text citations. Some have a comma after the authors names, some do not...

The figures all look fine, and I appreciate the diagram of the experimental arena.

Experimental design

The Methods section is the section that I believe needs to be greatly improved to be published in PeerJ.

-First, the authors need to describe the order in which the fish experienced the three tests (activity level, exploration, and boldness). Did all fish experience the tests in the same order? Also, if boldness was not the first test, I suspect that the fish may have had too much experience in the experimental arena for it to be considered a true boldness test of leaving a refuge (considering that the fish may already have information that they would be safe outside of the refuge). Please refer to Balaban-Feld et al. (2019) Influence of predation risk on individual spatial positioning and willingness to leave a safe refuge in a social benthic fish. Behav Ecol Sociobiol 72: 87.

- For the boldness test, did the authors only measure time spent in refuge versus time spent in open area. Did the authors not measure latency to leave the refuge? This would seem to be the most important indicator of boldness.

-Where exactly were the fish placed into the tank for the exploration test and the general activity test. Along the side of the tank? This needs to be explained in detail.

-Please provide a more detailed explanation of how repeatability was calculated.

-Finally, I believe the hypothesis and purpose of the study should be more clearly defined. For me, I only really understood the major finding of the study in the second to last paragraph of the discussion (lines 228-236)

Validity of the findings

I think that the study does have merit, in that it demonstrates how researchers may be able to skip overall activity testing and simply run exploration or boldness tests to find personality differences. I believe this result should be the result that is most clearly explained as it is the most important (lines 234-236).

I also believe the abstract should be heavily edited, it was very hard to follow and I think it could explain the experiment design more clearly, and spend less time focusing on the novel object result.

Furthermore, I am not sure I understand or agree with the author’s assertion that the fish did not care about the presence of a novel object. Although the authors found similar movement metrics between the exploration and general activity tests, they found that fish in the exploration experiment reached the novel object quicker than fish reached the ‘invisible object’…line 204-205 states “the fish purposefully encountered the object earlier for exploration”. Does this not clearly show that the fish did care about the presence of a novel object. So, if this is true, it may be possible for researchers to find important information about differences in exploratory behavior among fish individuals (even if swimming speeds and other metrics don’t change between contexts).

---

## Round 0.2 · Minor Revisions

I am sorry for the time it has taken to complete my review of your revised manuscript. The manuscript arrived at about the same time as several others, creating a backlog which was made worse by the holiday period.

Only one reviewer was available to comment on your revised manuscript. We agree that the manuscript has been substantially improved. Unfortunately, there are still some areas that have not been adequately resolved.

I agree with all the reviewer’s suggestions. Several deserve to be emphasized.
• In particular, I accept his point that distance to center is not a meaningful measure in a small, rectangular aquarium and can be removed without loss.
• I also agree that it is awkward to present your results in the Introduction. I have provided a suggestion for alternative text on the annotated manuscript.
• I agree that you should provide a specific justification for the choice of 1.75 cm/s as a threshold for movement.
• Several of the reviewer’s other comments note sections that he found unclear. For most of them, I have provided suggestions on the annotated manuscript. Please check carefully to be sure that I have not altered your intended meaning in each case.

I also have some concerns not mentioned by the reviewer.

Major concerns

Implications of study design: Now that you have made clear the design of your study, potential problems have become apparent. These have important potential implications for the strength of your conclusions and must be discussed objectively and carefully.
• Although you subjected the fish to three tests, you tested each fish on only a single occasion. This would normally not be sufficient to show that individuals expressed a consistent temperament or syndrome. It is a logical possibility that similarities between test are due to factors other than individual differences, such as the time of day, amount of time in the holding tank before testing, noise in the environment, stress of capture from the holding tank or others that you or I may not have thought of. This limits your conclusions and requires some careful thought about its implications. When you discuss the correlations (first section of the Discussion) it is important to point out that the correlations may be related to factors other than individual differences, what the implications are for your conclusions about a single test, and perhaps what sort of future tests would be required to show that activity in the boldness or exploration context is an adequate substitute for a separate activity test.
• Another problem is that you performed the three tests in the same order for all fish. This means that differences between the three tests could be a result of test order or time in the arena. For example, when the fish is first put into an aquarium, especially without any conspecifics, it may be stressed from being captured and moved, then try to escape, then explore its environment, then either tire or calm down. This might explain a reduction in swimming speed or percent time moving from boldness to exploration to activity context. Again, my suggestions only raise some of the possible confounding effects. The strict logic is that anything that changes with time in the aquarium could have influenced the differences between tests (either increasing or decreasing the differences). The implications of such potentially confounding effects should be included in the second part of the Discussion where you discuss the average differences between contexts.
• These explanations should not be too long, but they should be clear and informative and based on careful thought about the implications. If you wish to discuss them with me before resubmitting the manuscript to be sure that you understand my concerns and that you have adequately responded with clear wording, I invite you to contact me directly by email [email protected].

Distance to the novel object: You provide a calculation for distance to the novel object, but the formula seems to provide a value only for a single frame. How was the value actually calculated, for example the mean or median or minimum of all values or some other measure? In addition to explaining your calculation in the methods, whenever you refer to the distance to the novel object, add an adjective, e.g.,‘median distance’ to make the measure clearer. (This ambiguity also applies to the distance from the center of the arena, but may not be a problem if you eliminate this variable.)

Other concerns
There are many small errors in grammar or word use. I spent considerable time suggesting corrections on the annotated version. You may contact me directly about any that you don’t understand so that the next revision will be sufficient to have the paper accepted. Below, I use line numbers to emphasize some of the points, particularly those that are repeated.

L22. I think the preposition ‘in’ is preferable to ‘under’ for the contexts, e.g., “measured in a boldness context”. Please change here and throughout the manuscript.
L29. Latency is a measure based on time. Therefore, the word ‘time’ is redundant and should be removed (here and elsewhere in the manuscript).
L60-63. This goal implies that you wish to compare the values in each context with all other contexts. Please revise to make it clear that you only plan to compare boldness and exploration with activity but not with each other. In the paragraph developing your goals, make it clear why this would be your goal, rather than examining all comparisons.
L66-67 The wording about habitat use and sociality must be clarified. I was not able to understand what you meant about the use of open water and cover. Do not use the term ‘prefers’ unless preference has been tested in a choice experiment. I suggest something like ‘This omnivorous species often forms shoals in open water but may also be found solitarily, refuging in plants or behind stones when foraging or avoiding predators.’ (This sentence is only a guess at what you meant to express. Change the sentence to properly express the habitat use of this species.)
L86. I accept the use of SE for your experimental results, but you should use SD to describe the variability of your test subjects. It would be useful to also provide the range of sizes.
L87. I assume you mean that the stream of capture was a tributary of the Wujiang River.
L91,92,95. w is not a standard abbreviation; spell out ‘weeks’
L100. If you wish to refer to the tank as an arena, you must define it at first mention.
L148. Provide the justification for the threshold for swimming with a sentence here such as “We eliminated movements less than 1.75 cm because . . . . (reference).” It is important to provide the explanation at first mention. Then you can briefly refer to it without repeating the references for the other measures.
L189. You only need to describe the Wilcoxon test as nonparametric at the first mention. After that, just refer to the Wilcoxon test.
L219. For latency, your methods indicates use of Wilcoxon because data were not normally distributed, but the results indicates a t-test.
L303ff. There are still mistakes in the references.
Fig. 1. Part of caption is missing. The dimensions do not agree with the text (L100).
Fig. 2,4. There are spelling mistakes on the figure panels.

·

Basic reporting

Points detailed below in General comments, otherwise OK.

Experimental design

Points detailed below in General comments, otherwise OK.

Validity of the findings

Points detailed below in General comments, otherwise OK.

Additional comments

The manuscript is much improved and it is much clearer what the aim of the study is.

As I stated in my first review, the distance to the centre does not seem appropriate because the test arena is narrow (20cm, which is around 3 body lengths (L172)) – the centre of the arena is still very close to the nearest tank wall. As there weren’t any significant trends with this variable, it should be removed from the paper entirely. It just adds unnecessary text (e.g. L 230) so the story is less clear, and due to the tank size is probably not biologically of value anyway; there is nothing lost by removing it.

The abstract needs to start with a general sentence that sets the background and context for the present study which then justifies the next sentence starting “The aim of the present study”. Similarly, there should be a concluding sentence at the end that puts the study’s results back into this broader context.

L 31-33: Here it needs to be stated what direction the trend is in rather than just that the behaviour “significantly differed between the two contexts”. It also needs to be clearer why the results for this mean that this might be “a potential variable … in the future”; currently this is not clear.

L 38: Cite either just across-taxa reviews or a papers on non-fish too. This is a general biology journal, not a fish-specific one.

L 43: Cite an early paper to use the tendency to leave a refuge as a measure of boldness, not a recent one.

L 51: “behavioral syndrome” not “behavior syndrome”.

L 73: The results can’t be stated in the Introduction. Instead the paragraph could start with “To test further whether the novel object is a sufficient stimulus…”.

L 79-80: This is unclear.

L 102: “an opaque canvas to stimulus from the observers during the experiments” is unclear.

L 123: “the linking threat manipulated remotely” is unclear.

L 135: What evidence is there that 10 minutes is enough “to eliminate the effect of the exploration context”? How could this even be tested?

L 149: This line can be deleted as the information is already given earlier.

L 151: It is still not clear why a threshold “above 1.75 cm s-1” is used – both the references are from the author’s research group. There needs to be a proper justification in this paper rather than the reader having to search the cited papers for the details (presuming the justification is actually given in these papers, rather than being arbitrarily chosen).

L 153: Cite Ioannou et al. 2015 after “as opposed to goal-directed behavior in fish”.

L 187: It needs to be stated why it is just the median swimming speed for these pairwise comparisons.

L 188: It is nonparametric, not nonparameter.

L 240-248: This section needs rewriting. Firstly, it needs to be clear that the faster speeds and more time spent moving in the boldness test could be because the fish can rely on returning to the shelter when threatened, while when not available, they reduce activity (speed and time spent moving) because a refuge is not available (assuming this is the point being made). References need to be included here, such as
Herbert-Read, J. E., Kremer, L., Bruintjes, R., Radford, A. N., & Ioannou, C. C. (2017). Anthropogenic noise pollution from pile-driving disrupts the structure and dynamics of fish shoals. Proceedings of the Royal Society B: Biological Sciences, 284(1863), 20171627. https://doi.org/10.1098/rspb.2017.1627.
which show that activity is reduced when perceived threat increases.

The next point that “it might have little biological significance given the large difference among individuals.” needs fuller explanation. In terms of ecological impact, e.g. the effect the fish have on their prey, it may be true that inter-individual variation may be more important than whether the fish have a shelter or not. Again, references need to be included, i.e. those that look at the ecological consequences of personality variation, such as
Ecological consequences of the bold–shy continuum: the effect of predator boldness on prey risk
CC Ioannou, M Payne, J Krause Oecologia 157 (1), 177

Finally, it is surprising to see the point that “variables that have been frequently used in previous studies, such as the percentage of time spent outside of a shelter, might be reliable indicators of boldness in fish species” when these variables have not been included in the current study but could easily be measured from the videos and included in the analysis. Either the current study is only focused on movement behaviour outside a shelter (if present) or it is broader than that. This last part (L 245-247) should either be deleted or such analyses included in the paper and the broadness of the paper expanded.

L 266-273: Again, I found this section unclear. I would expect the
Herbert-Read JE, Krause S, Morrell LJ, Schaerf TM, Krause J, Ward AJW. 2013. The role of individuality in collective group movement. Proceeding Royal Society B 280: 20122564
paper to be cited here.

L 276: Unclear. Also it is surprising not to see mention of neophilia in this section of the Discussion. Or that the fish perceived the ‘novel’ object as something that was potentially rewarding, such as food. These are biologically plausible explanations for the result that the fish approach the novel object sooner than expected if it wasn’t present (a very strong effect, based on Fig. 5b) but these aren’t considered at present.

There are a number of inconsistencies in the references list e.g. spacing and punctuation in the volume and page numbers. Bevan et al. is listed here with no page numbers, and not cited in the main text (it can be cited on L 44), and is not on its own line in the reference list.

Figure 1: The height and width dimensions are not consistent with the Methods.

Christos Ioannou
University of Bristol

---

## Round 0.3 · accepted · Accept

I appreciate the careful corrections of the manuscript. It now reads very well, and I consider it ready for publication. Two small corrections are needed, however. For clarity on L203, change to 'positively correlated between contexts (Fig. 4)' for additional clarity. For the caption to Figure 1, add 'and E) novel object' and remove the 'and' before D.